# Extent of Unidentified Complaints and Depression Is Inversely Associated with Fish and Shellfish Intake in Young Japanese Women

**DOI:** 10.3390/nu17071252

**Published:** 2025-04-03

**Authors:** Toshikazu Suzuki, Yui Yoshizawa, Shiori Takano

**Affiliations:** 1Graduate School of Human Ecology, Wayo Women’s University, 2-3-1 Konodai, Ichikawa, Chiba 272-8533, Japan; 2Department of Health and Nutrition, Wayo Women’s University, 2-3-1 Konodai, Ichikawa, Chiba 272-8533, Japan; jieshengjize@gmail.com (Y.Y.); t.instinc.shio@gmail.com (S.T.)

**Keywords:** unidentified complaints, medically unexplained symptoms, depression, food frequency questionnaire, eicosapentaenoic acid, docosahexaenoic acid, vitamin B_12_, vitamin D

## Abstract

**Background/Objectives**: Vague physical complaints with no corresponding organic disease background are called unidentified complaints. The symptoms of patients with unidentified complaints closely resemble medically unexplained or persistent physical symptoms, with the onset sometimes masked by mental disorders. Over the past 50 years, numerous studies have connected unfavorable eating habits to these symptoms; however, no study has systematically examined the association between the symptoms and specific nutrients or food items. **Methods**: We conducted a cross-sectional study of young Japanese women, using questionnaire surveys, to assess their nutritional intake, quantify unidentified complaints and depression, and identify nutrients/food items primarily associated with the severity of these conditions. **Results**: Our findings indicate that participants with high scores for unidentified complaints, depression, or both had lower intake levels of eicosapentaenoic acid, docosahexaenoic acid, vitamin D, and vitamin B_12_ than those with low scores, alongside reduced fish and shellfish consumption. Notably, the median fish and shellfish intake in the group with high scores for both unidentified complaints and depression was less than one-fourth of that in the low-score group. **Conclusions**: The results align with previous findings, demonstrating a modest inverse association between fish intake and depression risk, and suggesting the involvement of fish and shellfish intake in the occurrence of unidentified complaints.

## 1. Introduction

Vague physical complaints with no corresponding organic disease background, including general malaise, easy-to-seize fatigue, lower-limb malaise, palpitations, shortness of breath, numbness in the hands and feet, stomach bloating, and brain fog, are called unidentified complaints [1]. These symptoms are also referred to as indefinite or unspecified complaints, general malaise, or unexplained physical symptoms [2,3,4,5] and closely align with medically unexplained symptoms and persistent physical (somatic) symptoms [6,7]. Abe of Toho University, Japan, first described unidentified complaints in the 1960s as beriberi-like symptoms without vitamin B_1_ deficiency [8]. He also classified the unidentified complaints into three types: the neurotic type, primarily caused by neurological factors without dysautonomic symptoms; the psychosomatic type, where mental disorders induce dysautonomic symptoms or physical and mental aspects of dysautonomia interact; and the essential dysautonomia (vegetative disorder) type, which is unrelated to psychogenic factors [1,8]. Women are more likely to report unidentified complaints than men [1,9]. The prevalence of these complaints increases during periods of hormonal changes, such as menopause, puberty, and sexual maturity [2,10,11,12]. Psychosocial stress is related to unidentified complaints [3,9,13], suggesting that an unhealthy mental state is highly associated with the symptoms of unidentified complaints. Additionally, various background factors related to unhealthy lifestyles, including life rhythm collapse, low sleep quality, skipping breakfast or other meals, and unbalanced diet, have been reported to be associated with unidentified complaints in various cross-sectional surveys for the last five decades [4,14,15,16,17,18,19,20,21,22], implying the involvement of nutrients in unidentified complaints. However, these studies seldom outline the relationship between unidentified complaints and food or nutrient intake.

Ishiwata, Hashizume, and their colleagues developed a 26-item questionnaire listing unidentified complaints commonly observed in individuals with suspected micronutrient deficiency. This tool aimed to screen candidates suspected of marginal micronutrient deficiency by calculating the complaint score [23,24]. They also used another questionnaire to screen for specific micronutrient deficiency individually. As a result, they also established a threshold of the complaint score for identifying potential micronutrient deficiency and found a weak-to-moderate correlation between the score and overall micronutrient intake deficiency [23,25]. However, specific nutrients and foods associated with unidentified complaint scores remain unspecified.

In this study, we conducted a cross-sectional survey of young Japanese women using the abovementioned unidentified complaints questionnaire [23], referred to here as the micronutrient deficiency-related complaints questionnaire (MDCQ), and a food frequency questionnaire on food groups (FFQg) [26,27]. The FFQg estimates individual nutrient and food intake based on the frequency of 29 food groups and 10 cooking methods answered over the previous 1–2 months, allowing us to identify nutrients or food items specifically correlated with high complaint status on the MDCQ. Additionally, since psychiatric symptoms are linked to unidentified complaints, we also used the Beck Depression Inventory Second Edition (BDI-II) [28,29], a widely used depression assessment tool, to examine correlations between nutrient or food intake and both depression severity and the combined depression–unidentified complaint score.

## 2. Materials and Methods

### 2.1. Study Design

This study was conducted in accordance with the Declaration of Helsinki guidelines and approved by the Wayo Women’s University Human Research Ethics Committee (No. 2304). A questionnaire survey was administered from June 2023 to December 2023. Written informed consent was obtained from all participants prior to their participation in the study. Eighty-six volunteers who belonged to Wayo Women’s University as undergraduate students (aged 18–27 years) were included, and no exclusion criteria were applied. Of the 86 students, 80 majored in nutrition, 4 in nursing, and 2 in home economics. The participants answered three questionnaires—the FFQg, MDCQ, and BDI-II—on the same day. The anthropometric parameters needed to answer the FFQg were not measured when participants knew their heights and weights. Height and weight were measured using a stadiometer (YG-200; Yagami Inc., Nagoya, Japan) and body scale (HBF-375; Omron Corporation, Kyoto, Japan), respectively, only when the participants did not know the exact value. Body mass index (BMI) was calculated by dividing body weight (kg) by height (m) squared.

### 2.2. Questionnaires

#### 2.2.1. FFQg

The FFQg, version 6, an optional software feature within Excel-Eiyo-kun^®^ version 9 (Kenpakusha, Tokyo, Japan), was used to estimate the average daily intake of nutrients and food items among Japanese individuals [26,27]. This software is compatible with the Dietary Reference Intakes for Japanese (2020) [30] and the Standard Tables of Food Composition in Japan—2015 (Seventh Revised Version) [31]. The questionnaire sheet can be printed using the software. The participants recalled their habitual diets over the previous month, selected portion sizes, and recorded the weekly use frequency of 29 food groups and 10 cooking methods. Standard portion sizes for each food item were provided in both written descriptions and color illustrations. The intake of dietary supplements, except protein, was not included in the questionnaire. Individual nutrient and food intake estimates were automatically calculated by inputting each participant’s data into the software.

#### 2.2.2. MDCQ

The questionnaire consists of 26 symptoms related to unidentified complaints suspected to be associated with micronutrient deficiency [23,25]. The participants recalled their symptoms over the previous week and rated their frequency using the following four categories: “never” (0 points), “sometimes” (1 point), “often” (2 points), and “always” (3 points). The questionnaire’s developers established the upper limit of a normal score as mean + 2 × (standard deviation) = 26, with scores ≥ 27 considered indicative of possible micronutrient deficiency. Therefore, participants scoring ≤26 were included in the low complaint (LC) group, while those scoring ≥27 were included in the high complaint (HC) group, indicating a potential micronutrient deficiency.

#### 2.2.3. BDI-II

The Japanese version of BDI-II, validated for both adults and undergraduate students in Japan, was used [29,32]. The questionnaire consists of 21 items, with participants recalling their mental state related to depression over the previous 2 weeks. Each item was rated on a four-point scale (0–3 points), with total scores ranging from 0 to 63. Based on these scores, the participants were classified as follows: ≤13 (minimal depression), 14–19 (mild depression), 20–28 (moderate depression), and ≥29 (severe depression) [29,32]. Therefore, participants scoring ≤13 were placed in the low depression (LD) group, while those scoring ≥14 were placed in the high depression (HD) group.

### 2.3. Data Analysis

Microsoft Excel for Microsoft 365 (Version 2501) (Microsoft Corporation, Redmond, WA, USA) and IBM SPSS Statistics (Version 28.0.1) (IBM Japan, Ltd., Tokyo, Japan) were used for statistical analysis. Results were expressed as median (minimum–maximum, min–max) or mean (standard deviation, SD). The Mann–Whitney U-test was used to assess differences between the two groups. Spearman’s rank correlation test and the chi-squared test were performed to analyze the correlation and association between MDCQ and BDI-II scores, respectively. A *p*-value < 0.05 was considered statistically significant.

## 3. Results

The anthropometric, nutrient, and food intake data obtained from the FFQg survey of the 86 participants were compared with those of Japanese women, cited from the National Health and Nutrition Survey Japan, 2019 (Table 1) [33]. The participants’ body compositions were similar to the average for Japanese women aged 20–29 years. Although direct comparisons were limited due to differences in dietary survey methods, we inferred that the participants’ nutritional intake was comparable to that of Japanese women in the 20–29 age group by drawing parallels between the datasets.

The distribution of MDCQ and BDI-II scores among the participants is shown in Figure 1. The mean (SD) and median (min-max) MDCQ scores were 19.7 (9.1) and 19 (4–41), respectively, while the BDI-II scores were 8.5 (6.7) and 6.5 (0–24). The mean MDCQ score was higher than that of a similar age group (18–30 years) in a previous study [14.0 (9.0)] [24]. Conversely, the mean BDI-II score was lower than that reported for 558 Japanese women undergraduate students [13.0 (9.0)] [32] and similar to that of U.S. undergraduate students [8.4 (7.2)] [34]. This suggests that, on average, the participants had minimal depressive symptoms; however, their complaint levels were slightly elevated compared to those of the general population of young adults in Japan.

The participants were categorized into two groups based on MDCQ scores (≤26: LC, ≥27: HC) or BDI-II scores (≤13: LD, ≥14: HD), and their nutritional and food intake levels were compared (Appendix A). BMI and intake of energy-yielding nutrients did not differ significantly between the LC and HC groups (Appendix A). However, BMI, protein intake, and grain consumption were significantly lower in the HD group than in the LD group (Appendix A). Additionally, energy and carbohydrate intake tended to be lower in the HD group, although not statistically significantly, suggesting a reduced appetite among HD participants. Regarding nutrients, eicosapentaenoic acid (EPA), docosahexaenoic acid (DHA), vitamin D, and vitamin B_12_ intake levels were statistically lower in both HC and HD groups than in the LC and LD groups (LC vs. HC: Figure 2, Appendix A; LD vs. HD: Figure 3, Appendix A). Fish is the primary dietary source of EPA and DHA [35]. It is also the primary dietary source of vitamin D and vitamin B_12_ for the Japanese population [36,37]. As expected, the median fish and shellfish intake in the HC group was two-thirds of that in the LC group (Figure 2e), with a similar reduction observed in the HD group compared to that in the LD group (Figure 3e). These findings indicate that the severity of unidentified complaints and depression was associated with reduced EPA, DHA, vitamin D, and vitamin B_12_ intake, primarily due to reduced fish and shellfish consumption.

Since symptoms of unidentified complaints partially overlap with those of depression [1], we assessed the correlation between MDCQ and BDI-II scores using Spearman’s rank correlation coefficient. The *ρ* value of for the correlation between these scores was 0.577 (*p* = 5.94 × 10^−9^), indicating a moderate correlation. A chi-squared test was further conducted to determine the association between the HC and HD groups (Figure 4). Of the 21 participants in the HC group, 13 (61.9%) were also in the HD group, whereas only 6 of the 59 LC participants (9.2%) were in the HD group (*p* = 6.15 × 10^−7^). These findings suggest that HC participants were more likely to experience high depression severity, and those belonging to both HC and HD groups exhibited severe unidentified complaint symptoms.

The participants were further divided into LC-LD (≤26 MDCQ and ≤13 BDI-II) and HC-HD (≥27 MDCQ and ≥14 BDI-II). The anthropometric parameters and daily nutrient and food intake levels between the LC-LD and HC-HD groups were compared (Figure 5, Appendix A). Similarly to previous group comparisons, EPA, DHA, vitamin D, and vitamin B_12_ intake levels were significantly lower in the HC-HD group than in the LC-LD group. Additionally, the median EPA and DHA intake levels in the HC-HD group were less than half of those in the LC-LD group, while vitamin D and B_12_ intake levels were 40% lower. Furthermore, the HC-HD group exhibited a significant reduction in other micronutrients, including zinc, selenium, molybdenum, niacin, and pantothenic acid. The median fish and shellfish intake level in the HC-HD group was 75% lower than that in the LC-LD group (Figure 5e), suggesting that fish and shellfish consumption may play a role in preventing the progression of unidentified complaints and depression.

## 4. Discussion

In this study, we found that the level of fish and shellfish consumption was associated with the severity of unidentified complaints and depression status. Additionally, EPA, DHA, vitamin D, and vitamin B_12_ intake levels were lower in the HC and HD groups than in the LC and LD groups, respectively. We also observed a moderate correlation between MDCQ and BDI-II scores, supporting previous findings that unidentified somatic symptoms and depression overlap, but do not fully coincide [1,38]. Moreover, the HC-HD group, characterized by high complaint levels and high depression status, consumed only one-fourth of the fish and shellfish intake of the LC-LD group, who were considered relatively healthy both physically and mentally. Furthermore, the estimated intake levels of vitamin D and *n*-3 polyunsaturated fatty acids in the HC-HD group were below adequate levels (vitamin D: 8.5 µg/d; *n*-3 polyunsaturated fatty acids: 1.6 g/d for women aged 18–29), while vitamin B_12_ intake levels were close to their recommended dietary allowance (2.4 µg/d) for the same age group [30]. Therefore, individuals with high scores for unidentified complaints and depressive symptoms were likely to have insufficient intake levels of very long-chain *n*-3 fatty acids (EPA and DHA) and vitamin D.

Several studies over the last 25 years have reported a modest inverse association between fish consumption and depression risk. Two meta-analyses concluded that the pooled relative risks of depression for the highest vs. lowest fish consumption were 0.83 (95% confidence interval [CI]: 0.74–0.93) and 0.89 (95% CI: 0.80–0.99), respectively [39,40]. Fish is a rich source of protein and very-long-chain *n*-3 fatty acids (EPA and DHA), along with essential minerals (selenium, potassium, magnesium, iodine, zinc) and vitamins (vitamin D, vitamin B_12_) [41]. Supplementation with EPA, DHA, and the abovementioned micronutrients has been considered an adjunctive strategy in antidepressant treatment [42,43]. Recent randomized controlled trials have demonstrated that EPA- and DHA-rich fish oil supplementation enhances Mediterranean-style dietary interventions for mental health improvement, and vitamin D supplementation serves as an effective adjunct to antidepressant therapy [44,45]. Additionally, Mendelian randomization studies have identified the role of *n*-3 polyunsaturated fatty acids in depression etiology and of vitamin D in dementia etiology, suggesting their contributions to brain structure and mental health [46,47]. Furthermore, bioactive peptides from fish protein hydrolysates exhibit antioxidative, neuroprotective, and anti-inflammatory properties, which may also contribute to health benefits in combination with EPA and DHA [48]. Therefore, shortages of EPA, DHA, vitamin D, and bioactive peptides caused by habitually low or no fish intake may increase the risk of unidentified complaints as well as of psychogenic or depressive symptoms.

We noted that protein, zinc, selenium, molybdenum, niacin, and pantothenic acid intakes were lower in the HC-HD group than in the LC-LD group (Appendix A). Of these nutrients with different intakes between the groups, only zinc did not fulfill the recommended dietary allowance (8 µg/d) [30]. Because dietary zinc intake was inversely associated with mood disorders, including depression and anxiety in the cross-sectional study in the Iranian female students, we cannot exclude the possibility that less zinc intake might also be involved in unidentified complaints with depressive symptoms [49]. In addition, FFQg can only estimate essential nutrients, not non-essential nutrients, such as the so-called phytochemicals, including polyphenols, whose supplementation could alleviate depression and anxiety [50]. Therefore, we cannot exclude the association between the dietary intake of such compounds and the extent of unidentified complaints or depression.

When Ishiwata and Hashizume developed the MDCQ questionnaire, they aimed to screen for latent deficiencies of multiple micronutrients, including vitamins A, D, E, B_1_, B_2_, B_12_, and C; pantothenic acid; folic acid; calcium; magnesium; iron; zinc; and copper [23,25]. However, only vitamins D and B_12_ were significantly lower in the HC and HD groups than in the LC and LD groups, respectively. When both questionnaire (MDCQ and BDI-II) scores were considered together, only pantothenic acid and zinc were additionally identified as being lower in the HC-HD group than in the LC-LD group. The lack of significant findings for other micronutrients could be explained by the low prevalence of complaints related to their deficiencies. For example, in a previous study, fewer than one-sixth of outpatients suspected of vitamin B_1_ deficiency due to unidentified complaints or neurological symptoms were biochemically diagnosed with authentic or latent vitamin B_1_ deficiency (whole blood vitamin B_1_ < 27 ng/mL) [51]. Moreover, vitamin B_1_ is widely present in various animal and plant-based foods, potentially masking vitamin B_1_ insufficiency in dietary assessments. In contrast, up to 70% of young Japanese women have been diagnosed with vitamin D deficiency (serum 25-hydroxyvitamin D <20 ng/mL), with habitual low fish intake closely linked to this deficiency [52].

This study had some limitations. First, we used the food frequency method, FFQg, to estimate daily nutrient and food group intake. As a result, the estimated nutrient and food amounts are approximate rather than absolute. The dietary record method, which requires participants to record foods or dishes or take photos of each meal for several consecutive days, provides more precise quantitative estimates. However, this method demands significant effort from both participants and researchers, making it unsuitable for surveys lasting more than 3 days. In contrast, the food frequency method is effective for ranking participants based on their nutrient or food group intake [26]. Additionally, this study primarily included nutrition students, who have a relatively high level of knowledge about nutritional surveys. Although it might underestimate the differences in fish and shellfish intake compared to general population, as the students could be aware of the benefits of these food components, we considered it reasonable to use the FFQg as an initial screening tool to identify nutrient and/or food shortages in intake that may be associated with the severity of unidentified complaints, depression, or both combined. As a next step, targeting a wide range of adult women with a larger group, including students other than those majoring in nutrition, dietary records focusing specifically on fish intake, and habitual supplement consumption, may provide a more precise estimation of EPA, DHA, and vitamin D intake, facilitating more quantitative investigations into their role in preventing or managing unidentified complaints with mental health symptoms. Second, we did not conduct a biochemical nutritional assessment. Although randomized controlled trials have demonstrated that fish consumption increases serum levels of EPA, DHA, and 25-hydroxyvitamin D [53,54], direct measurements of serum EPA, DHA, and 25-hydroxyvitamin D levels are required to accurately analyze the relationship between nutritional status and the severity of unidentified complaints and depression. Therefore, future studies should quantify these serum levels in addition to estimating their dietary intake.

Third, we did not investigate the possibility of covariates or cofounders in the analysis. One possible such factor could be socioeconomic status, which is involved both in the quality of dietary intake [55] and the prevalence of depressive symptoms [56] in Japanese people. Therefore, household incomes should also be considered in further investigations. Fourth, this study could not establish whether habitually low fish intake causes or increases the risk of unidentified complaints and depression. Medically unexplained symptoms, resembling unidentified complaints, are caused by immune-inflammatory activation triggered partly by oxidative stress [57]. Numerous studies have reported that EPA, DHA, and vitamin D exhibit anti-inflammatory properties [58,59]. Consequently, it is reasonable to consider EPA, DHA, vitamin D, and fish as nutrients or foods that may contribute to preventing or managing unidentified complaints and depression, as suggested in a recent review by Raza et al. [60]. However, multiple factors beyond nutritional deficiencies influence the development of unidentified complaints and depression. Further standardized and robust intervention studies are required to evaluate the efficacy of fish consumption and/or EPA, DHA, and vitamin D intake in enhancing neuropsychiatric health.

## 5. Conclusions

The intake of EPA, DHA, vitamin D, and vitamin B_12_, primarily derived from fish and shellfish, was lower in participants with high MDCQ and BDI-II scores, indicating the severity of unidentified complaints and depression, than in those with low scores. When combining both surveys, fish and shellfish intake in the HC-HD group was less than one-fourth of that in the LC-LD group. The median intake levels of *n*-3 polyunsaturated fatty acids and vitamin D in the HC-HD group were below the adequate intake levels for Japanese women aged 18–29 years, while vitamin B_12_ intake level was close to the recommended dietary allowance. These findings align with the results of previous studies demonstrating a modest inverse association between fish intake and depression risk. Further research is needed to clarify the efficacy of fish consumption and/or EPA, DHA, and vitamin D intake in preventing and managing neuropsychiatric-related unidentified complaints and depression.

## Figures and Tables

**Figure 1 nutrients-17-01252-f001:**
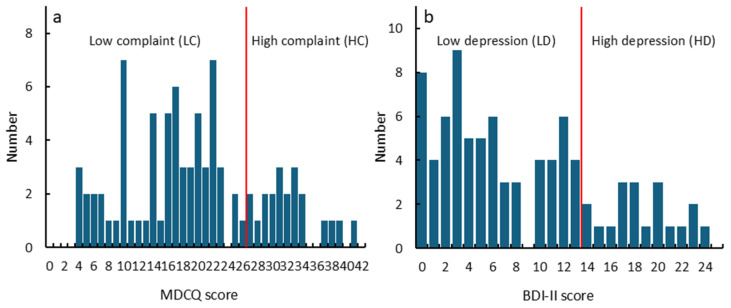
Distribution of MDCQ (**a**) and BDI-II (**b**) scores among participants. Red lines indicate the cutoff between low- and high-score groups.

**Figure 2 nutrients-17-01252-f002:**
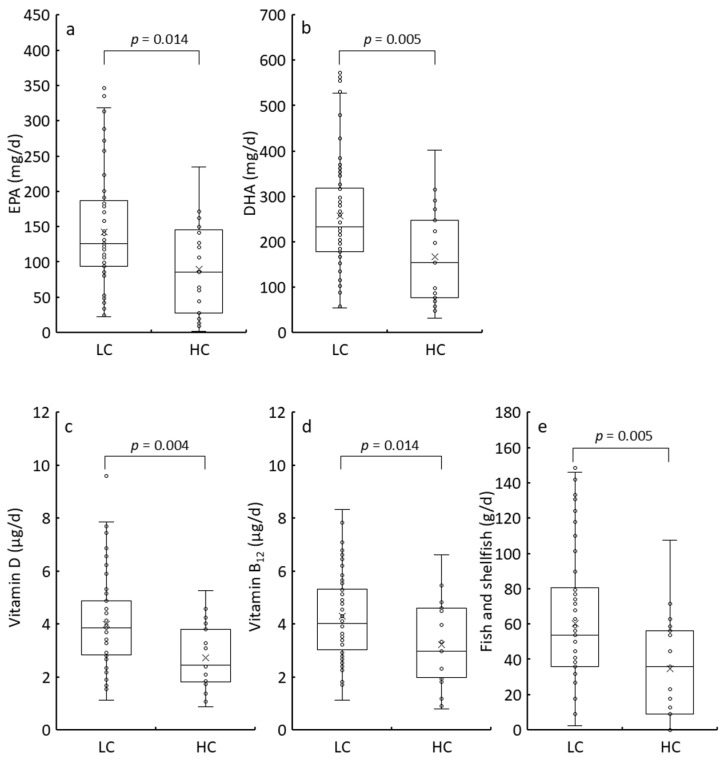
Comparison of EPA (**a**), DHA (**b**), vitamin D (**c**), vitamin B_12_ (**d**), and fish and shellfish (**e**) intake between the LC (≤26 of MDCQ score) and HC (≥27 of MDCQ score) groups by box plots. The bottom of the box is the 25th percentile, the line that intersects the box is the median, the multiplication sign within the box is the mean, and the top of the box is the 75th percentile. Whiskers above and below the box represent the 10th and 90th percentiles, and the points above and below the whiskers indicate the outliers. The *p*-values from the Mann–Whitney U-test for the LC and HC groups are displayed in the top square brackets. EPA, eicosapentaenoic acid; DHA, docosahexaenoic acid; MDCQ, micronutrient deficiency-related complaints questionnaire; LC, low complaint; HC, high complaint.

**Figure 3 nutrients-17-01252-f003:**
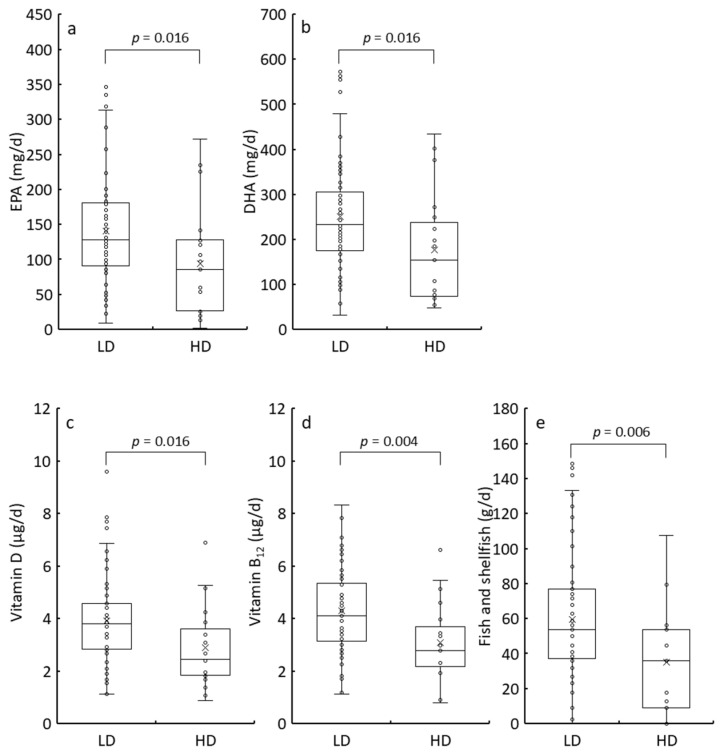
Comparison of EPA (**a**), DHA (**b**), vitamin D (**c**), vitamin B_12_ (**d**), and fish and shellfish (**e**) intake between the LD (≤13 of BDI-II score) and HD (≥14 of BDI-II score) groups by box plots. The bottom of the box is the 25th percentile, the line that intersects the box is the median, the multiplication sign within the box is the mean, and the top of the box is the 75th percentile. Whiskers above and below the box represent the 10th and 90th percentiles, and the points above and below the whiskers indicate the outliers. The *p*-values from the Mann–Whitney U-test for the LD and HD groups are displayed in the top square brackets. EPA, eicosapentaenoic acid; DHA, docosahexaenoic acid; LD, low depression; HD, high depression; BDI-II, Beck Depression Inventory, Second Edition.

**Figure 4 nutrients-17-01252-f004:**
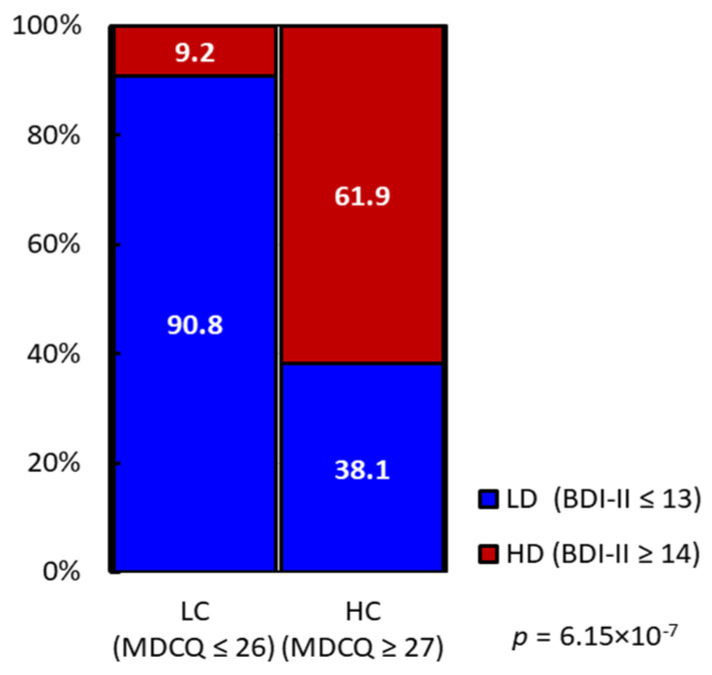
Frequency of participants in the HD (≥14 of BDI-II score) and LD (≤13 of BDI-II score) groups within the HC (≥27 of MDCQ score) and LC (≤26 of MDCQ score) groups. The frequency of HD associated with HC and the *p*-value from the chi-square test results are shown. BDI-II, Beck Depression Inventory, Second Edition; MDCQ, micronutrient deficiency-related complaints questionnaire; LC, low complaint; HC, high complaint; LD, low depression; HD, high depression.

**Figure 5 nutrients-17-01252-f005:**
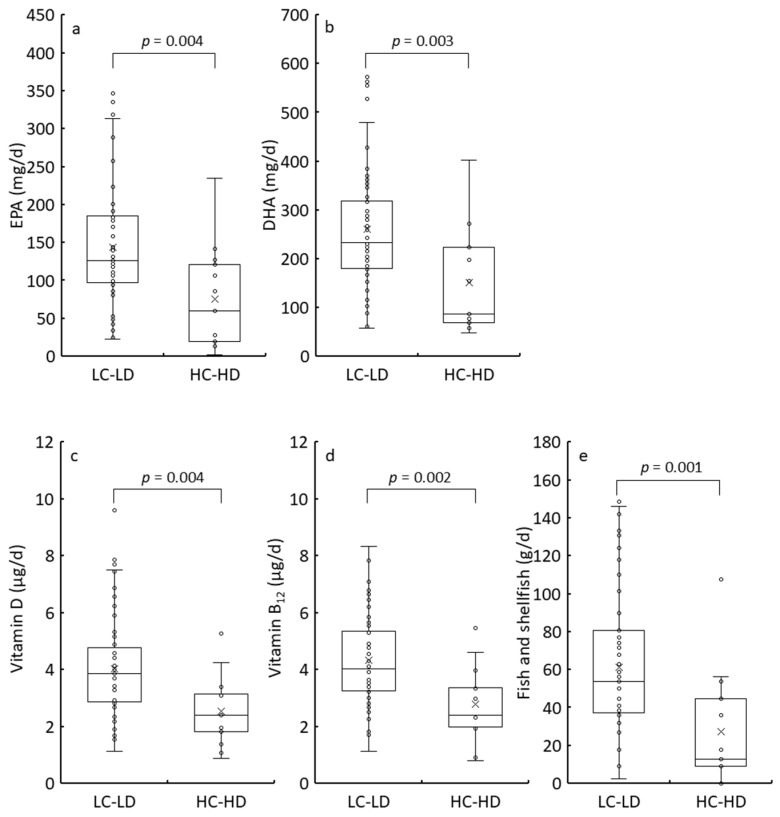
Comparison of EPA (**a**), DHA (**b**), vitamin D (**c**), vitamin B_12_ (**d**), and fish and shellfish (**e**) intake between the LC-LD (≤26 of MDCQ score and ≤13 of BDI-II score) and HC-HD (≥27 of MDCQ score and ≥14 of BDI-II score) groups by box plots. The bottom of the box is the 25th percentile, the line that intersects the box is the median, the multiplication sign within the box is the mean, and the top of the box is the 75th percentile. Whiskers above and below the box represent the 10th and 90th percentiles, and the points above and below the whiskers indicate the outliers. The *p*-values from the Mann–Whitney U-test for the LC-LD and HC-HD groups are displayed on the top square brackets. EPA, eicosapentaenoic acid; DHA, docosahexaenoic acid; MDCQ, micronutrient deficiency-related complaints questionnaire; BDI-II, Beck Depression Inventory, Second Edition; LC, low complaint; HC, high complaint; LD, low depression; HD, high depression.

**Table 1 nutrients-17-01252-t001:** Baseline characteristics and estimated nutrient intake of participants.

Parameter	All (*n* = 86)		Japanese Women Aged 20–29 ^†^
	Median (Min–Max)	Average (SD)	Average (SD)
Age (years)	20 (18–27)	20.1 (1.1)	-
Women/men (*n*/*n*)	86/0	-	-
Anthropometrics			
Height (m)	1.60 (1.45–1.73)	1.59 (0.05)	1.58 (0.06)
Weight (kg)	51.5 (35.2–75.0)	52.2 (6.3)	52.0 (8.1)
BMI (kg/m^2^)	20.4 (15.2–30.7)	20.7 (2.4)	21.0 (2.9)
Daily nutrient intake			
Energy (kJ)	7289 (2247–11067)	7159 (1569)	6694 (1861)
Carbohydrate (g)	231 (70–393)	226 (52)	202 (64)
Protein (g)	60.0 (20.5–111.5)	59.1 (14.7)	61.1 (18.4)
Fat (g)	59.4 (18.9–111.2)	59.0 (15.6)	55.5 (21.9)
(Fatty acids)			
Saturated fatty acid (g)	19.0 (6.2–35.9)	19.0 (5.8)	17.1 (8.1)
Monounsaturated fatty acid (g)	21.2 (6.8–40.0)	20.9 (5.7)	20.7 (8.8)
*n*-6 polyunsaturated fatty acid (g)	10.0 (3.2–23.2)	10.5 (3.4)	9.1 (4.6)
*n*-3 polyunsaturated fatty acid (g)	1.82 (0.53–4.66)	1.94 (0.67)	1.82 (1.20)
α-linoleic acid (mg)	1339 (436–3599)	1479 (568)	NR
Eicosapentaenoic acid (EPA) (mg)	122 (2–346)	130 (80)	NR
Docosahexaenoic acid (DHA) (mg)	220 (32–573)	236 (128)	NR
(Minerals)			
Sodium (g)	2.93 (1.14–6.68)	3.06 (1.12)	3.28 (1.23)
Potassium (g)	2.06 (0.80–4.15)	2.03 (0.64)	1.74 (0.66)
Calcium (mg)	416 (151–897)	435 (153)	408 (210)
Magnesium (mg)	216 (86–389)	212 (63)	192 (72)
Iron (mg)	6.5 (2.5–11.3)	6.4 (1.9)	6.2 (2.5)
Zinc (mg)	7.3 (2.5–13.8)	7.1 (1.8)	7.3 (2.7)
Copper (mg)	1.0 (0.4–1.5)	1.0 (0.3)	0.90 (0.32)
Manganese (mg)	2.4 (1.0–4.0)	2.4 (0.7)	NR
Iodine (µg)	478 (14–3259)	637 (535)	NR
Selenium (µg)	52.3 (17.5–89.3)	53.5 (13.8)	NR
Chromium (µg)	7.1 (2.9–12.7)	6.8 (2.2)	NR
Molybdenum (µg)	153 (45–297)	154 (50)	NR
(Vitamins)			
Vitamin A (µgRAE ^‡^)	386 (118–880)	394 (163)	447 (878)
Vitamin D (µg)	3.5 (0.9 -9.6)	3.7 (1.7)	4.6 (5.9)
Vitamin E (α-tocopherol) (µg)	6.0 (2.3–13.6)	6.3 (1.9)	5.4 (2.8)
Vitamin K (µg)	193 (67–405)	197 (84)	207 (179)
Vitamin B_1_ (mg)	0.93 (0.41–2.12)	0.92 (0.26)	0.77 (0.37)
Vitamin B_2_ (mg)	1.00 (0.36–1.95)	1.00 (0.29)	0.97 (0.43)
Niacin (mgNE ^‡^)	25.1 (9.7–50.8)	24.9 (6.8)	25.6 (9.2)
Pantothenic acid (mg)	5.0 (2.4–9.8)	5.0 (1.3)	4.65 (1.73)
Vitamin B_6_ (mg)	1.0 (0.4–2.3)	1.0 (0.3)	0.91 (0.38)
Biotin (µg)	25.8 (11.0 -46.2)	26.8 (8.2)	NR
Folic acid (µg)	231 (94–466)	234 (85)	226 (129)
Vitamin B_12_ (µg)	3.9 (0.8–8.3)	4.0 (1.7)	4.3 (4.2)
Vitamin C (mg)	64.3 (14.2–166.5)	67.1 (31.5)	62 (48)
Daily food intake			
Grains (g)	672 (73–1431)	633 (170)	352 (142)
Potatoes (g)	25.0 (0–131.2)	31.9 (28.3)	35.4 (48.8)
Green-yellow vegetables (g)	15.9 (1.9–52.1)	18.1 (10.9)	58.8 (60.8)
Light-colored vegetables (g)	29.6 (0–90.2)	32.0 (19.2)	137.8 (109.2)
Fruits (g)	22.4 (0–117.7)	32.6 (29.8)	52.7 (95.3)
Legumes (g)	58.7 (0–264.1)	67.2 (49.8)	48.1 (69.3)
Fish and shellfish (g)	53.8 (0–148.7)	54.3 (36.0)	41.6 (49.9)
Meat and poultry (g)	189 (25–669)	201 (99)	108.6 (72.5)
Eggs (g)	43.1 (0–129.4)	46.9 (20.9)	34.4 (34.7)
Dairy foods (g)	135 (13–612)	148 (98)	104.5 (134.8)
Nuts (g)	2.6 (0–48.6)	7.8 (11.8)	1.3 (3.8)

Data are expressed as median (min–max) or average (SD) except for sex ratio. ^†^ Data were quoted from National Health and Nutrition Survey Japan, 2019 [33]. NR denotes data not reported. ^‡^ RAE, retinol activity equivalents; NE, niacin equivalents.

## Data Availability

The original contributions presented in this study are included in the article/Appendix A. Further inquiries can be directed to the corresponding author.

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
