# Peer review of "Extent of Unidentified Complaints and Depression Is Inversely Associated with Fish and Shellfish Intake in Young Japanese Women"

_nutrients, 2025, doi:10.3390/nu17071252_

Round 1
Reviewer 1 Report
Comments and Suggestions for Authors
The MDCQ questionnaire should be more specifically explained in the body of the text including what the acronym stands for. A few examples of symptoms should be provided here. Not all readers will be familiar with this questionnaire.
There is no mention of covariates or possible confounders in the analysis. If the authors did not determine that controlling for confounders was necessary then this should be specifically stated with rationale.
Reviewer 2 Report
Comments and Suggestions for Authors
Suzuki and colleagues provide evidence showing that the extent of unidentified complaints and depression is inversely associated with fish and shellfish intake in young Japanese women. The Author concludes that their results align with previous findings demonstrating a modest inverse association between fish intake and depression risk, and suggesting the involvement of fish and shellfish intake in the occurrence of unidentified complaints. Despite the manuscript is well-written and interesting, some points should be addressed.
- The Authors should better discuss the role of biological sex in the context of mood disorders. In this respect, mood disorders are more common in women than in men (PMID: 30061743).
- The Authors must a check and correct statements without references, adding the appropriate references.
- The Authors should better enlarge the discussion about the beneficial effects of a correct diet on depressive symptoms. The Authors might want to report and discuss the following papers (PMID: 36829831; PMID: 34819888 and others).
- The Authors must check and correct possible typos throughout the manuscript.
Reviewer 3 Report
Comments and Suggestions for Authors
The manuscript accounts for an interesting study showing an inverse association of unidentified complaints and depression with fish and shellfish intake in young Japanese female students. While the problem of relationship between depression and fish consumption was the subject of numerous studies and metaanalyses, the question of unidentified complaints has attracted less attention.
The study was well designed, presentation of results and discussion are proper. Discussion covers limitations of the study. Basing the study mainly on students majored in nutrition might provide a higher reliability of food frequency questionnaire and even underestimate the differences in fish and shellfish uptake with respect to the general population as the students could be aware of benefits of these food components; a study exceeding beyond this group could provide even more distinct results.
In the Discussion, the Authors concentrate on EPA, DHA, and vitamin D. Inspection of Supplementary materials shows that also protein, grain and zinc intake was lower in the HC-HD group, as compared with the LC-LD group. May it cause deficiency in essential amino acids? The effect of zinc on the mental state has been also well demonstrated. The Authors could consider covering these questions in the Discussion.
The size of the group is somewhat limited. A follow-up study on a larger group, not limited to students, and covering other age groups, would be valuable.
Conclusions are scientifically sound.
Minor remarks:
Lines 37, 54, 269: Please delete the scientific degrees and titles.
Table 1 and Supplementary: Please correct “selen” to “selenium”
